# Cysteine Restriction in Murine L929 Fibroblasts as an Alternative Strategy to Methionine Restriction in Cancer Therapy

**DOI:** 10.3390/ijms222111630

**Published:** 2021-10-27

**Authors:** Werner Schmitz, Elena Ries, Corinna Koderer, Maximilian Friedrich Völter, Anna Chiara Wünsch, Mohamed El-Mesery, Kyra Frackmann, Alexander Christian Kübler, Christian Linz, Axel Seher

**Affiliations:** 1Department of Biochemistry and Molecular Biology, Biocenter, D-97074 Wuerzburg, Germany; wschmitz@biozentrum.uni-wuerzburg.de; 2Department of Oral and Maxillofacial Plastic Surgery, University Hospital Wuerzburg, D-97070 Wuerzburg, Germany; ries.elena@icloud.com (E.R.); corinna.koderer@posteo.de (C.K.); Max.Voelter@gmx.de (M.F.V.); wuenschanna@outlook.de (A.C.W.); kyra.frackmann@t-online.de (K.F.); Kuebler_A@ukw.de (A.C.K.); LINZ_C@ukw.de (C.L.); 3Department of Biochemistry, Faculty of Pharmacy, Mansoura University, Mansoura 35516, Egypt; m_elmesery@mans.edu.eg

**Keywords:** methionine restriction, cysteine restriction, mass spectrometry, LC/MS, cancer therapy, caloric restriction, homocysteine, amino acid analogues, cysteine synthase inhibitor

## Abstract

Methionine restriction (MetR) is an efficient method of amino acid restriction (AR) in cells and organisms that induces low energy metabolism (LEM) similar to caloric restriction (CR). The implementation of MetR as a therapy for cancer or other diseases is not simple since the elimination of a single amino acid in the diet is difficult. However, the in vivo turnover rate of cysteine is usually higher than the rate of intake through food. For this reason, every cell can enzymatically synthesize cysteine from methionine, which enables the use of specific enzymatic inhibitors. In this work, we analysed the potential of cysteine restriction (CysR) in the murine cell line L929. This study determined metabolic fingerprints using mass spectrometry (LC/MS). The profiles were compared with profiles created in an earlier work under MetR. The study was supplemented by proliferation studies using D-amino acid analogues and inhibitors of intracellular cysteine synthesis. CysR showed a proliferation inhibition potential comparable to that of MetR. However, the metabolic footprints differed significantly and showed that CysR does not induce classic LEM at the metabolic level. Nevertheless, CysR offers great potential as an alternative for decisive interventions in general and tumour metabolism at the metabolic level.

## 1. Introduction

In recent years, caloric restriction (CR) has established itself as a common option both for the prevention and for the treatment of age-associated diseases of the cardiovascular system, type II diabetes, and cancer [1,2]. CR means that an organism continuously consumes 10–30% less energy than the usual daily intake (approximately 2000 kcal). Initially, CR was assessed as an experimental phenomenon in selected species, such as *Drosophila melanogaster* and *Caenorhabditis elegans*, which led to a significant and drastic increase in their absolute lifespans. However, the results were also reproduced in mice and rats [3]. The effects of this complex phenomenon can be explained in detail at the cellular and even molecular levels.

From an extremely simple point of view, the cell knows two states: dividing or not dividing. The ability to divide depends on two key factors: energy and mass. Both factors are determined within the cell at every point in time at various locations and via various mechanisms [4,5]. If even one factor is not sufficiently available over a longer period of time, the cells switch to a mechanism that we call low-energy metabolism (LEM). 

LEM is implemented by a molecular network in which the protein kinase mTOR (mechanistic target of rapamycin) plays a decisive role [6]. If mTOR is activated, it can massively promote growth and proliferation. If mTOR is inhibited, proliferation and growth are suppressed or completely inhibited. Among other things, the process of autophagy is initiated, which enables the digestion, recycling, and regeneration of cellular components as well as molecules, such as amino acids [7].

In the case of energy, sirtuins and AMPK (AMP-activated protein kinase) are two of the most important sensors. A high AMP/ATP quotient signals a lack of ATP and activates AMPK, which in turn inhibits mTOR. Sirtuins measure another significant unit of energy—NAD(+). A high NAD(+) level also signals a low energy level and activates sirtuins, which ultimately leads to the inhibition of mTOR again [8]. 

In the case of mass, amino acids are one decisive factor in proliferation. Most of the newly generated mass of a proliferating cell is not formed by glucose or even lipids but by amino acids taken up extracellularly [4]. For this reason, mTOR is also a sensor for the mass. Selected amino acids (e.g., leucine, arginine, glutamine, serine, and methionine) are regularly quantified by mTOR [9]. 

It is not clear why the abovementioned amino acids are monitored by mTOR, but there are probably three reasons for detecting methionine. (1) If you summate the individual reaction steps necessary for the synthesis of methionine, methionine is the most energy-intensive amino acid and is therefore a very valuable resource [10]. (2) The S atom of the sulfhydryl group of methionine is essential in metabolic processes that are sulfur-based. One example is the synthesis of cysteine from serine. In this case, methionine provides the S atom of the sulfhydryl group. (3) Methionine also plays a central role in many synthetic steps. S-adenosylmethionine (SAM) is one key product. It is formed from methionine through reaction with ATP. SAM functions in metabolism as a methyl donor, whereby it is hydrolysed to adenosine and homocysteine via S-adenosyl homocysteine. Homocysteine can be remethylated again to methionine or broken down to the amino acid cysteine [11].

As a rule, mTOR does not measure amino acids directly but rather via amino acid sensors, which then form larger protein complexes with mTOR and thus influence its activity. In the case of methionine, SAMTOR plays this role, binding directly to GATOR1 and thus inhibiting mTORC1. However, SAMTOR does not bind methionine but, as the name suggests, SAM. Methionine is measured only indirectly via the SAM level [12]. The absence of amino acids like methionine leads, like the lack of energy, to the inhibition of mTOR and thus to the inhibition of growth and proliferation. For this reason, not only CR but also protein restriction (PR) and amino acid restriction (AR) are effective and lead to more or less the same positive effects via almost an identical mechanism to CR [13,14].

How can AR be implemented in vitro and in vivo? A method of simulating methionine restriction (MetR) has been established in the laboratory. With the help of media that lack methionine, it can be easily implemented in cell culture. MetR can also be implemented quite easily in animal models due to the composition of the fed pellets [15,16]. The situation is different if MetR is to be used, for example, in cancer therapy. Lowering the amount of methionine in food is not easy. One trick is the use of L-methionine-γ-lyase (methionase), a bacterial enzyme that catalyses the conversion of L-methionine, which can be taken directly with food. This use has been successful both in vitro and in vivo [17]. However, are there more efficient alternatives to the use of methionase and to lowering the in vivo methionine level?

One option is the already mentioned close metabolic connection between methionine and cysteine. Cysteine restriction (CysR) could therefore lead to an endogenous decrease in methionine. However, the problem here is exactly the same: cysteine must be removed from nutrition. The synthesis of cysteine offers a point of attack. The amount of cysteine consumed in the body is higher than the amount ingested through food. In addition, cysteine is often increasingly used in the cell for the de novo synthesis of glutathione [18]. If endogenous cysteine synthesis is now inhibited, this could, on the one hand, lead to the conversion of methionine into cysteine precursors (e.g., homocysteine or cystathionine), which then cannot be further processed into cysteine but lower the methionine level. Furthermore, the lack of cysteine per se could lead to a reaction similar to MetR: the induction of LEM. Semiessential cysteine is synthesized from serine and homocysteine, the source of which is mainly essential methionine. In the first step, the two metabolites condense to cystathionine under catalysis by cystathionine-β-synthase (CBS) and are then cleaved to cysteine and α-ketobutyrate by deamination by cystathionine-γ-lyase (CSE). In this case, already known inhibitors of the enzymes CBS and/or CSE necessary for cysteine synthesis offer a therapeutic approach. Successful use against tumour cells has already been demonstrated both in vitro and in vivo [19]. 

Another possibility are amino acid analogues (AAAs). In the last 10 years, the relevance of amino acids in tumour development and thus the use of AAA in tumour therapy has come to the fore [20,21]. In theory, analogues of methionine or cysteine can certainly induce LEM via competition.

In earlier work, we used the murine cell line L929 as a model system and analysed the influence of MetR on metabolism using high-resolution liquid chromatography/mass spectrometry (LC/MS). L929 cells react rapidly and efficiently to MetR, and an analysis of more than 150 different metabolites belonging to different classes (amino acids, urea and TCA cycles, carbohydrates, etc.) by LC/MS defined a metabolic fingerprint and enabled the identification of specific metabolites reflecting normal or MetR (LEM) conditions [22].

In this work, we analysed the influence of CysR on the murine cell line L929. The focus was on whether CysR, like MetR, can induce LEM. For this purpose, the proliferation behaviour of L929 under Met(-) and Cys(-), both individually and in combination, was analysed over a period of five days. In addition, a metabolic analysis using LC/MS was carried out every 24 h over a period of five days. The metabolic footprint was compared with that created under MetR. In the first experiments, the effects of different methionine and cysteine analogues on the proliferation behaviour of L929 cells was also investigated. In addition, the effects of already known cysteine synthesis inhibitors on the proliferation behaviour of L929 cells were analysed.

## 2. Results

A typical feature of MetR is the mTOR-mediated inhibition of proliferation. In the murine cell line L929, MetR leads to efficient inhibition in a relatively short period of time (24–48 h). In the first analysis here, the proliferative behaviour was analysed under CysR.

### 2.1. Cysteine Restriction Is an Alternative and Efficient Way to Inhibit L929 Cell Proliferation

For this purpose, cells were incubated for five days with complete medium (Met+/Cys+), cysteine-(Met+/Cys-), or methionine-free (Met-/Cys+) medium, or, for the control, in medium missing both amino acids (Met-/Cys-). After one, three, and five days, the cells were analysed via crystal violet staining, and the relative cell numbers were determined (Figure 1a).

After just 24 h, cells showed inhibition of proliferation for all types of ARs, which increased continuously up to day 5 (Figure 1a). Even if the combination (Met-/Cys-) had the strongest effect, the inhibitory effects of individual treatments were more or less at the same level. Since crystal violet staining enables the analysis of only the relative cell numbers, cells were analysed in a further experiment under the same conditions using digital microscopy. By staining the nucleus with Hoechst 33342, the cells were automatically counted in a defined area. This enables the analysis of the absolute cell numbers and determines how the total cell number changes under MetR or CysR, i.e., whether the cells decelerate proliferation or whether the cells die (Figure 1b). After 24 h, the cells were still proliferating under all conditions. While the maximum cell number was reached after 72 h in the control, the number of cells remained constant under restrictive conditions; no cells obviously died (Figure 1b).

### 2.2. Homocysteine and S-Adenosylmethionine Compensate for MetR but Not CysR

Methionine is an essential amino acid, while cysteine is defined as a semiessential amino acid because it can be synthesized from serine and the essential amino acid methionine. In principle, both amino acids cannot be synthesized de novo in mammals. Needs must be met through food intake. However, it is possible to resynthesize both amino acids from various preliminary and intermediate stages. Typical intermediates are SAM or homocysteine. However, tumour cells and immortalized cells often show so-called methionine dependency. These cells have lost the ability to regenerate methionine and consequently cysteine from intermediates and are dependent on extracellular supply [23,24]. For this reason, the L929 cell line was analysed for its potential to compensate for the missing amino acids using the precursors DL-homocysteine, SAM, and D- and L-homoserine under MetR and CysR by self-synthesis. Interestingly, only under MetR can homocysteine compensate for the restricted amino acid, SAM, albeit to a slightly lesser extent (Figure 1c). Cysteine was not restored to its level in the control by either homocysteine or SAM (Figure 1d).

### 2.3. Metabolic Fingerprinting of Cysteine Restriction in L929

To generate metabolic profiles, L929 cells were examined under CysR over a period of 120 h. Cell pellets and supernatant were obtained in triplicate every 24 h and analysed by LC/MS. In the following, the results of two independent experiments are summarized in metabolic groups and presented with a focus on key areas. An overview of all the results is attached in the Appendix A.

#### 2.3.1. CysR Influences the Total Mass of Essential Metabolic Pathways

A simple and quick way of analysing mass spectrometric results is to group them together. These findings are clarified by using the sum of the metabolite masses associated with these pathways. In addition to the meaningfulness of the relative concentration of individual molecules, the summed relative masses of all relevant metabolites belonging to one pathway (e.g., the TCA cycle) provide relevant information about its overall regulation. This type of analysis was performed for amino acids (except cysteine), the urea cycle, the TCA cycle, carbohydrates, pyrimidines, and purines in both cell extracts (Figure 2a) and media (Figure 2b).

A summary of the groups shows intracellular results, which are presented in the following detailed examination of the individual metabolites. Namely, at almost every point in time, the concentrations of most metabolites under CysR were higher than those in the control. Since CysR inhibited proliferation (Figure 1a,b) and reduced metabolism is assumed, a large number of metabolites seem to accumulate within the cell. While there were smaller differences for amino acids and the urea cycle, the differences in the other groups were even clearer. The TCA cycle deserves special mention. Under metabolically active conditions, this is one of the most stressful cycles. Precisely because its intermediates are precursors for a large number of metabolites (e.g., amino acids and glucose), the cycle is also of enormous relevance for cancer cells [25]. Under CysR, the concentrations were increased by approximately two-fold compared to those of the control (Figure 2a). In the medium, on the other hand, the profiles are quite similar for most groups. For the urea cycle, slightly increased amounts were seen in the control in the first 96 h. The most striking difference was seen in the amino acids. The amino acid content in the medium continuously decreased in the control, i.e., the cells took up amino acids, but the total concentration under CysR remained relatively constant (Figure 2b).

#### 2.3.2. The LC/MS Profile under CysR for Different Categories of Metabolites

The metabolic profile for amino acids showed a very similar intracellular picture for the essential amino acids (Figure 3a) both in the control and under CysR. After 72 h, the highest concentrations, close to 100%, were reached under both conditions and then fell back to the 24 h starting value after 120 h. Among the nonessential amino acids, cysteine showed the most striking difference. While the intracellular content in the complete medium increased up to 100%, under CysR, it increased from only 2% to 12%. Notably, the highest values under CysR were found for the nonessential amino acids, and the contents of individual amino acids were higher than those of the control even after 96 and 120 h. In the supernatant, there was a somewhat more differential profile for the essential amino acids (Figure 3b). Most of the amino acids were absorbed in the control and, accordingly, usually showed the lowest concentration after 120 h. Under CysR, on the other hand, the concentration hardly changed over time or even increased for some amino acids. For the nonessential amino acids, the courses over time were similar and mostly different in only intensity. Cysteine and glutamine were exceptions for the supernatant under CysR, with concentrations decreasing to much lesser extents, as was proline, whose concentration increased significantly more.

The methionine salvage pathway (MSP) recycles sulfur-containing metabolites to the amino acid methionine. This pathway is found in all types of organisms from unicellular bacteria to plants and animals and starts with 5′-methylthioadenosine (MTA). The main source of MTA is a byproduct in the synthesis of polyamines, and it originates from SAM, which is formed by condensing methionine and ATP [26]. In Figure 4a, metabolites that are direct metabolites of or correlate with MSP are shown. This group is of interest because it contains molecules that serve as precursors for cysteine synthesis. Alpha-keto-glutaramate, SAH, and decarboxy-SAM, which are present in significantly higher concentrations over the entire period under CysR, are noticeable here. The polyamines putrescine and spermidines, which serve as markers for cells with high metabolic activity [27,28], show very similar courses under the two conditions. However, the concentration of N-Ac-putrescine under CysR was already very low after 24 h compared to that of the control.

The profiles for molecules classified into the class of sulfur-containing “S” metabolites were also quite similar under the two conditions (Figure 4b). Drastic differences can be seen for cystine, the reduced compound GSH, and the oxidized form of glutathione (GSSG), whose concentrations were extremely lower under CysR. Interestingly, a precursor of GSH, glutamylcysteine, is present at a lower concentration in CysR in the first 48 h but then increases after 120 h to almost the maximum value. Under both conditions, intermediate and precursors of cysteine occur in relatively high amounts. Formally, the formation of cysteine does not mean a transformation but the depletion of methionine. Homocysteine is a precursor where conversion to both methionine and cysteine is relatively easy. Cystathionine results from the condensation of homocysteine and serine and from subsequent hydrolysis to cysteine and homoserine, which is further broken down to alpha-ketobutyrate. An attempt to compensate for cysteine deficiency can be seen in the first 48 h, since the cystathionine content under CysR was almost 100% and the homocysteine content was higher than that in the control even after 96 and 120 h.

The next category was glycine metabolism (Figure 4c). However, the focus here is not directly on glycine but on its role in the synthesis of creatine, which is composed of the amino acids glycine, arginine, and methionine. Creatinine is the breakdown product of creatine and is formed irreversibly without enzymatic catalysis in aqueous solution. Sarcosine is a byproduct and an intermediate product in glycine synthesis or degradation. All three named metabolites showed significantly higher concentrations of approximately 100% after 72 h under CysR. The reverse is true for the metabolite methylglycine, which is a precursor to sarcosine.

In the glycolysis group, the focus was on the classic metabolites of this pathway. One of the first steps is the phosphorylation of glucose and thus its activation. Under CysR conditions, the hexose-P concentration was continuously higher than that of the control. Classically, glycolysis is often divided into two sections: the processing of hexose into two trioses, from which energy is then obtained in the second part as ATP. Interestingly, this partition is reflected in the heat map of the glycolysis metabolites (Figure 4d). In the control, the concentrations of metabolites, including gyceral/glycerol phosphate, were significantly lower than those under CysR. What is particularly noticeable is the very high concentration of 100% after 24 h, which then decreased continuously over the course of time but always remained above the level of the control. In the second section, the concentration was very high under both sets of conditions, but the concentration was somewhat higher at almost every point in time compared to that of the control.

For purines and nicotinamides (and energy equivalents), only selected metabolites are presented (Figure 4e). Basically, a fairly uniform picture emerges here, since the highest values were found at almost every point in time under the CysR condition. ADP, ATP, GDP, and GTP showed very high values overall and, compared to those of the control, also generally higher values after 24 h. ATP and GTP decreased again after 24 h and reached an even slightly higher level than that of the control. The difference was very clear with the nicotinamides. Here, the values under CysR were generally higher (approximately two-fold on average) and rose continuously up to maximum values between 95% and 100% after 120 h.

### 2.4. CysR Initiates a Metabolic Profile Extremely Different from MetR

As already published by us, MetR efficiently results in a characteristic metabolic fingerprint and footprint in L929 cells [22]. The footprint represents a small group of metabolites, which allows a specific definition of AR. Under MetR, these metabolites were ADP/ATP, acetoacetate, creatine, spermidine, GSSG, UDP-glucose, and pantothenate (Figure 5a). All of the molecules mentioned show extreme differences in concentration compared to those of the control and were characteristic of LEM under MetR. Comparing the CysR footprint created in this work with the MetR footprint clearly shows that no characteristic LEM was induced under CysR (Figure 5b). While under MetR the concentrations of the metabolites mostly decreased sharply, most of the metabolites had significantly higher concentrations (or approximately the same concentration) under CysR than in the control. The only exception was GSSG, which was drastically reduced in both ARs. Two of the characteristic LEM markers, the low content of ATP (or ADP) and the decreasing or low content of spermidine, did not occur under CysR. Additionally, increased acetoacetate synthesis was observed under only MetR.

### 2.5. Simple Amino Acid Analogues for Methionine or Cysteine Are Inefficient for Inhibition of Proliferation in L929

CysR was investigated more closely because it is a possible alternative to methionine. One method for both amino acids is competition through various D-forms of the molecules, which leads to a displacement of the biologically active L-forms of the amino acids intracellularly and, in the best case, functions like an AR. The analysis of proliferation served as a read out since the inhibition of proliferation should be a characteristic of potential LEM. D-methionine, L-ethionine, D-ethionine, D-cysteine, formyl-L-methionine (F-L-methionine), and F_moc_-L-methionine were analysed (Figure 6a–g). Ethionine, which is very similar in structure to methionine, has already been successfully used in studies as a methionine analogue [29]. However, none of the D-forms used showed a clear biologically relevant effect. A slight but unconvincing effect was observed with only L-ethionine (Figure 6b). The use of Fmoc-L-methionine shows that the approach can work in principle. This molecule is used in the solid-phase synthesis of peptides. Fmoc (fluorenylmethoxycarbonyl) is an N-terminal-protecting group that enables selective peptide synthesis towards the C-terminus. The protective group prevents the reaction with the amino group of methionine. For this reason, Fmoc-methionine cannot be used, for example, in the extension of protein synthesis, since no N-terminal peptide bond, only a C-terminus bond, can be formed. Fmoc shows a clear antiproliferative effect (Figure 6f). For further analysis, Fmoc was analysed in a dilution series to show that the reaction was not unspecific because it was saturated (Figure 6g).

Basically, AR is difficult to implement as a therapy. For food intake, the amino acid in question can be removed only with difficulty and usually only reduced (and in a not clearly specific and standardized range). As already mentioned, cysteine offers the possibility of suppressing the synthesis of the amino acid from methionine by inhibiting the enzymes CBS and/or CSE. Numerous inhibitors have already been described in the literature [19]. In this work, we decided to analyse hydroxylamine-O-acetic acid (HOAA), hydroxylamine solution (HAS), DL-propargylglycine (DL-PG), and beta-cyano-L-alanine (beta-CA) to test the potential of the substances to inhibit proliferation in L929 cells (Figure 7a–d). HOAA and HAS were effective inhibitors with IC_50_ values of approximately 500 and 300 µM. However, HAS was most effective, as proliferation was reduced to up to 20%. DL-PG and β-CA showed only slight antiproliferative effects at very high concentrations.

## 3. Discussion

In recent years, MetR has established itself as a suitable method to investigate LEM induced by AR. Both in vitro and in vivo, MetR can activate biological and cellular reactions corresponding to CR [1]. The advantages and positive aspects of CR and AR are so strong that implementation as a (adjuvant) form of therapy, especially for cancer, appears logical and imperative. However, implementation as therapy is not easy. With CR, a great deal of discipline in regulating food intake is required over a long period of time. The difficulty is to remove/reduce the corresponding amino acids from the diet as much as possible. For this reason, alternatives are necessary that offer the possibility of either inducing an LEM at another level or limiting the corresponding amino acid. Caloric restriction mimetics (CRMs) offer a wide range of possibilities to intervene in metabolism and to induce CR. However, their potential has not yet been adequately investigated, and they also harbour the risk of side effects. Examples are metformin, which in recent years has been gaining in importance as a CRM, especially in cancer therapy Additionally, rapamycin would be an ideal candidate due to its strong inhibitory effects on mTOR; however, it has strong immunosuppressive effects [2].

In an earlier work, we showed that in the murine cell line L929, MetR induces a characteristic metabolic profile that corresponds to the expectations of LEM. In addition to a so-called metabolic fingerprint, which includes a relatively large number of metabolites, it was also possible to define a metabolic footprint that included both new and classic molecules that are decisive in CR or AR. For this reason, we examined the potential of CysR in the murine cell line L929. In addition to AR, cysteine also offers another point of attack. Cysteine consumption in the body is high and higher than the daily dose ingested through food. For this reason, cysteine is transported in high concentrations in the blood. In healthy subjects, cysteine is the most abundant plasma aminothiol (total concentration 250 µM). Approximately 65% is protein bound, 30% is free and oxidized, and 3–4% is reduced [30]. In addition, cells have the ability to synthesize cysteine from methionine. Cysteine is formed when methionine is broken down. GSH synthesis is one of the main reasons the cell has high cysteine requirements. As an antioxidant, GSH is absolutely necessary to maintain balance, and with its high potential of −240 mV, GSH is an ideal redox buffer [31]. At the same time, GSH also provides a cysteine reserve since cysteine can be released again when GSH is broken down. Human blood plasma contains approximately 1 mM cysteine in the form of total GSH [32]. For the reasons mentioned, a reduction in the cysteine concentration or a restriction in cysteine production have a massive influence on the metabolism of a cell. The advantage compared to methionine lies in the option of limiting in vivo cysteine synthesis by using appropriate inhibitors, so a simple drug approach is definitely possible.

The antiproliferative potentials of cysteine and methionine were compared in the first experiments. The ability to inhibit proliferation under restrictive conditions is absolutely necessary for potential cancer therapeutic approaches. Even the combination of both restrictions does not lead to any massive improvements, so the restriction of the individual amino acids is already close to the maximum inhibition of proliferation (Figure 1a). Analysis of the cells by digital microscopy also showed that the cells did not die under MetR and CysR, since the total number of living cells did not fall below the value measured at the start time (0 h) over the time period investigated (Figure 1b).

In the next experiment, we analysed the potentials of homocysteine and SAM to compensate for the corresponding AR. In some cases, tumour cells offer the advantage that they lose the ability to regenerate methionine from homocysteine and thus become more vulnerable [23,33]. The reverse synthesis of methionine from cysteine/homoserine in the human organism is described in the literature as fundamentally not possible [18]. The results show that homocysteine can 100% compensate for MetR. SAM, on the other hand, cannot fully compensate for MetR. This is astonishing since homocysteine can be metabolized from SAM via S-adenosylhomocysteine in two synthesis steps. The result for cysteine looks completely different. CysR was not compensated for by even the direct precursor homocysteine. The exact reason for this remains part of further investigations, but we postulate that the reactions or synthesis rates of the two enzymes CBE and CSE can be a reason for this. Basically, cystine provides the cells with a permanent source of cysteine via the blood supply. The reaction rates of the two enzymes mentioned may be sufficient to compensate for any deficiency, e.g., in GSH synthesis. However, as seen from the experiments, the ability to synthesize cysteine does not mean that in principle, the cell's total requirement for cysteine can be met.

The results of the experiment described above also offer an explanation for the mass spectrometry results. We do not want to detail the metabolic groups already described but would rather concentrate on the profile comparisons under MetR. An overview of all the results for CysR is attached as a Appendix A. The induction of LEM was very clear under MetR. Notably, almost all metabolites were present at extremely different concentrations. If they increased in the control, they decreased in MetR and vice versa. After 120 h, there were many metabolites at extremely different concentrations [22]. This is one of greatest differences when comparing MetR and CysR. Most noticeably, under CysR, the vast majority of the metabolites were present at a higher concentration than in the control. In addition, the concentrations of numerous metabolites remained constant for 120 h (until the end of the experiment) or even increased to their maximum values. A fundamental problem when interpreting the results is the simple mass spectrometric analysis. In contrast to flow analyses with isotope-marked metabolites, the analysis carried out determines the concentrations of a large number of metabolites, but the values are ultimately a snapshot, i.e., static. The results cannot be used to determine how much of the products have been converted in the meantime. A high concentration of a metabolite can, on the one hand, mean that a great amount has been synthesized. Conversely, a low concentration does not automatically mean that less has been synthesized but simply that correspondingly more has been consumed.

However, we assume with our results that the metabolic profile under CysR represents a metabolic standstill or a build-up of intermediate products rather than increased metabolism and synthesis rates due to the inhibition of proliferation. Any other interpretation would be difficult to justify since the cells no longer divide and require fewer metabolites than the control. Because the substrates are used up accordingly in the control, this is how the obviously high concentrations arise. This was particularly noticeable in the glycolysis group (Figure 4d) and for the energy metabolites ADP, ATP, GMP, GDP, GTP, NAD+, NADH, NADP+, and NADPH (Figure 4e).

However, the characteristic profile of CysR can be recognized very well from selected metabolites, which at the same time also reveals the potential of CysR as a potential therapeutic approach. Compared to that of the control, the cysteine concentration dropped dramatically to 2% after just 24 h. The concentration rose continuously up to 12% at 120 h. The cell can therefore generate cysteine but not completely compensate for its reduction. The same picture emerged for cystine, whose concentration dropped drastically, but the concentration remained almost constant over the period of the experiment. Correspondingly, the concentrations of GSH and GSSG also dropped dramatically. Most likely, less GSH can be synthesized, and in return, GSH is broken down to a greater extent to regenerate cysteine. Evidence for this could be the increasing taurine concentration under CysR, which is formed as a breakdown product of cysteine and whose concentration thus reflects intracellular cysteine turnover. The metabolites SAH, homocysteine, and cystathionine, which are relevant in cysteine synthesis, are also of interest. The SAH concentration was significantly higher under CysR, so the cell seemed to react to the reduced cysteine concentration by generating SAH, which could then be converted to cysteine via homocysteine and cystathionine. However, the last two metabolites mentioned showed similar concentration profiles over the test period both in the control and under CysR. These are probably the reaction steps that determine the speed of the reaction and that are already used to their maximum potential in the control due to proliferation. Accordingly, no more can be synthesized under CysR than in the control, and the missing cysteine can no longer be compensated for. This is also in agreement with the homocysteine experiment (Figure 1c,d). Analogously, SAH might not have the same effect as homocysteine itself since the synthesis steps to homocysteine are limiting but the conversion of homocysteine into methionine is not.

Basically, among the results that we obtained, CysR seems to have essentially an antiproliferative effect, the mechanism of which must be clarified in further experiments. LEM, with effects comparable to CR or MetR, is not induced at the metabolic level. This becomes clear when the footprints under MetR and CysR are compared (Figure 5a,b). Except for GSSG, no metabolite of the footprint corresponds to LEM. This is particularly clear for ADP, ATP, and spermidine, which are very good markers for energy-rich cells [27,28].

However, the crucial question is whether CysR is potentially suitable as a therapeutic approach even though LEM is not induced at the metabolic level. From our point of view, the answer is a resounding yes. First, CysR is antiproliferative and therefore effective. Second, the most important finding of this work is that, below a certain concentration, the low levels of cysteine compounds are no longer compensated for by the cells through their own synthesis. This means that a reduction in the cysteine concentration in vitro or in vivo below a certain threshold value could already have antiproliferative effects and that complete restriction is not mandatory. These low levels cannot be compensated for by precursors, such as homocysteine or SAH, which gives this method a clear advantage over MetR because in vivo cysteine can compensate for reduced methionine in some circumstances, e.g., in GSH synthesis [18,34]. Another advantage is the ability to intervene in cysteine synthesis through inhibitors. In this work, the inhibitors HOAA and HAS showed at least their antiproliferative potential in L929 cells. Other possible approaches would also be to inhibit cystine synthesis [19] and possibly the uptake of cystine via membrane transporters [35]. In addition, the methionine dependency often present in cancer cells automatically leads to an increased cysteine dependence, which could further increase the efficiency of CysR in vivo. Even if LEM is not induced, based on this work, CysR offers promising approaches to be pursued further as a potential therapeutic approach.

## 4. Material and Methods

### 4.1. Cell Culture

The murine fibroblast cell line L929 was purchased from the Leibniz Institute, DSMZ-German Collection of Microorganisms and Cell Cultures GmbH (Braunschweig, Germany). The cells were cultured in RPMI 1640 medium (Gibco, Life Technologies; Darmstadt, Germany) with 10% FCS (Sigma-Aldrich, Darmstadt, Germany) and 1% penicillin/streptomycin (P/S; 100 U/mL penicillin and 100 µg/mL streptomycin (Thermo Fisher Scientific, Darmstadt, Germany) at 37 °C in a humidified atmosphere containing 5% CO_2_. The base medium lacked methionine and cysteine/cystine. For full medium (named control, Met+ or Cys+), 15 mg/L-methionine (Sigma-Aldrich, Darmstadt, Germany) and 65.2 mg/L L-cysteine (Merck, Darmstadt, Germany) were added. For Met- medium, only cysteine was added, and for Cys-, only methionine was added (both at the concentrations mentioned above). For Met-/Cys-, no amino acid was supplemented.

### 4.2. Crystal Violet Staining (CytoTox Assay)

Cells were seeded at 10,000 cells in 100 µL of culture medium per well of a 96-well plate and incubated overnight. The following day, the cells were incubated in complete, methionine-free, cysteine-free, or methionine- and cysteine-free media. For compensation experiments, DL-homocysteine (Hcy), S-adenosylmethionine (SAM), or D-/L-homoserine (Hse) (all Sigma-Aldrich, Darmstadt, Germany) were used at 800 µM in Met- or Cys- medium. For competition experiments, (a) D-methionine (100×), (b) L-ethionine (100×), (c) D-ethionine (100×), (d) D-cysteine (50×), (e) formyl-L-methionine (F-L-methionine) (100×), and (f,g) Fmoc-L-methionine (100×) were used as indicated in 50× or 100× excess compared to the content in control media (15 mg/L L-methionine). A control was full medium, and a second control was the solvent at the same concentration as in the probe with the corresponding D-amino acid or analogue. For analysis of the cysteine synthesis inhibitor HOAA, DL-propargylglycine (DL-PG), hydroxylamine solution (HAS), and beta-cyano-L-alanine (beta-CA) (all Sigma-Aldrich, Darmstadt, Germany) were used from stock solutions in water with an assay starting concentration of 10 mM in a log2-diluted solution. For a control, full RPMI medium was used. The number of measured values, the incubation period, and the number of repetitions of the experiments are mentioned in the corresponding figure legend. For staining, the supernatants were removed, and the cells in each well were incubated with 50 µL of crystal violet solution (1% crystal violet in % methanol; Carl Roth, Karlsruhe, Germany) for 10 min and subsequently washed five times with distilled water. The plates were dried for 2 h in the dark. For quantification, 100 µL of methanol were added to each well, and the plate was incubated for 10 min until the crystal violet was completely dissolved. The photometric absorbance was measured at 595 nm using a microplate reader (Tecan, Crailsheim, Germany). For data analysis, the experiments were repeated at the indicated times to calculate the mean values and standard deviations. The results were normalized to the untreated control (100%). The relative cell number values determined via the crystal violet assay with the stimulated probes (CV_S_) were normalized to those of the untreated control (CV_C_) ((CV_S_/CV_C_) = CV_R_). To obtain percentage values, the CV_R_ value was multiplied by 100 (RCN (%) = (CV_S_/CV_C_) × 100 = CV_R_%).

### 4.3. ImageXpress Pico Automated Cell Imaging System—Digital Microscopy (Pico Assay)

Cells were seeded at 10,000 cells in 100 µL of culture medium per well of a 96-well plate and incubated overnight. The following day, the cells were incubated in complete, methionine-free, cysteine-free, or methionine- and cysteine-free media. The incubation time is stated in the corresponding figure legend. For staining, 10 µL of Hoechst staining solution (1:200 dilution in Hoechst 33342 medium (Thermo Fisher, Darmstadt, Germany) (10 mg/ml in H_2_O)) were added to each well, and the samples were analysed at this time point. After a 20–30-min incubation, wells were analysed with an ImageXpress Pico Automated Cell Imaging System (Molecular Devices, San Jose, CA, USA) via automated digital microscopy. The cells were analysed with transmitted light and in the DAPI channel at 4x magnification. The complete area of every well was screened. Focus and exposure time were set via autosetup and controlled by analysing 3–4 test wells. Finally, every result was confirmed visually and 95% of cells were counted and analysed.

### 4.4. L929 Experiments for Liquid Chromatography/Mass Spectrometry

L929 cells were seeded in 20 mL of medium in 15 cm Petri dishes. To prevent confluence during the test period, 1 × 10^6^ cells/Petri dish were seeded for day 1, 2, and 3 and 5 × 10^5^ cells for day 4 and 5 in triplicate. The media used for stimulation were prepared from a methionine-, cysteine-, and glutamine-free RPMI medium (Sigma-Aldrich, Darmstadt, Germany). The complete medium (control) contained 15 mg/L methionine, 65 mg/mL cysteine and 300 mg/L glutamine, the cysteine-free medium 15 mg/L methionine, and 300 mg/mL glutamine (amino acids from Sigma-Aldrich, Darmstadt, Germany). All media contained additional 10% FCS (Sigma-Aldrich, Darmstadt, Germany) and 1% penicillin/streptomycin (P/S; 100 U/mL penicillin and 100 µg/mL streptomycin (Thermo Fisher Scientific, Darmstadt, Germany)). After seeding, the cells were incubated on the following day with 20 mL of complete medium or 20 mL of cysteine-free medium per dish. Before harvesting, 1 mL of the supernatant was stored for analysis. The remaining medium was then removed, the cells were washed with 10 mL PBS, and detached with 3 mL of trypsin/EDTA (Thermo Fisher Scientific, Darmstadt, Germany). After adding 7 mL of the appropriate medium, the absolute cell number in the suspensions was analysed with the automated cell counter EVE^TM^ (NanoEntek (VWR, Darmstadt, Germany)). In order to obtain an accurate result, each sample was measured four times and the mean value was calculated. Pellets with 1 × 10^6^ cells were produced by centrifugation (5 min at 1200 rpm at RT). Until the LC/MS analysis, all samples were stored at −20 °C. The experiment was carried out twice (*n* = 2) and the results are summarized in the tables. 

### 4.5. Liquid Chromatography/Mass Spectrometry

Analysis of water-soluble metabolites in cell extracts and culture media.

Cells: After the addition of 0.5 mL of MeOH/CH_3_CN/H_2_O (50/30/20, *v*/*v*/*v*) containing 10 µM lamivudine, cell pellets were homogenized by ultrasound treatment (10∗1 s, 250 W output energy). Media: In total, 100 µL of culture medium were combined with 0.4 mL of MeOH/CH_3_CN (50/30, *v*/*v*) containing 10 µM lamivudine. The external standard lamivudine was not used for absolute metabolite quantification but used as a quality control in order to compensate eventually occurring technical issues. As quality control and for the determination of the corresponding retention times, most of the annotated metabolites (which are commercially available) were run as mixtures of pure compounds under identical experimental conditions. General procedure: The resulting suspension was centrifuged (20 kRCF for 2 min in an Eppendorf centrifuge 5424), and the supernatant was applied to a C18-SPE column that was activated with 0.5 mL of CH_3_CN and equilibrated with 0.5 mL of MeOH/CH_3_CN/H_2_O (50/30/20, *v*/*v*/*v*). The SPE eluate was evaporated in a vacuum concentrator. The resulting pellet was dissolved in 50 (cell extracts) or 500 µL (media extracts) of 5 mM NH_4_OAc in CH_3_CN/(25%/75%, *v*/*v*).

LC parameters: Mobile phase A consisted of 5 mM NH_4_OAc in CH_3_CN/H_2_O (5/95, *v*/*v*), and mobile phase B consisted of 5 mM NH_4_OAc in CH_3_CN/H_2_O (95/5, *v*/*v*).

After the application of 3 µL of the sample to a ZIC-HILIC column (at 30 °C), the LC gradient programme was as follows: 100% solvent B for 2 min, a linear decrease to 40% solvent B over 16 min, maintenance at 40% solvent B for 9 min, and an increase to 100% solvent B over 1 min. The column was maintained at 100% solvent B for 5 min for column equilibration before each injection. The flow rate was maintained at 200 μL/min. The eluent was directed to the ESI source of the QE-MS from 1.85 to 20.0 min after sample injection.

The MS parameters were as follows: scan type, full MS in the positive-and-negative mode (alternating); scan range, 69–1000 m/z; resolution, 70,000; AGC-target, 3E6; maximum injection time, 200 ms; sheath gas, 30; auxiliary gas, 10; sweep gas, 3; spray voltage, 3.6 (positive mode) or 2.5 kV (negative mode); capillary temperature, 320 °C; S-lens RF level, 55.0; and auxiliary gas heater temperature, 120 °C. Annotation and data evaluation: Peaks corresponding to the calculated monoisotopic masses (MIM +/- H^+^ ± 2 mMU) were integrated using TraceFinder software (Thermo Scientific, Bremen, Germany). Materials: Ultrapure water was obtained from a Millipore water purification system (Milli-Q Merck Millipore, Darmstadt, Germany). HPLC–MS solvents, LC–MS NH_4_OAc, and lamivudine were purchased from Merck (Darmstadt, Germany). RP18-SPE Columns: 50 mg of Strata C18-E (55 µm) in 1-mL tubes (Phenomenex, Aschaffenburg, Germany). Sonifier: Branson Ultrasonics 250 equipped with a 13-mm sonotrode (Thermo Scientific, Bremen, Germany).

LC/MS system: A Thermo Scientific Dionex UltiMate 3000 UHPLC system hyphenated with a Q Exactive mass spectrometer (QE-MS) equipped with a HESI probe (Thermo Scientific, Bremen, Germany). The samples were analysed with a high-resolution mass spectrometer, allowing the generation of XIC data, which were analysed by applying a very narrow m/z margin (+/− 3 mMU). Particle filter: Javelin filter with an ID of 2.1 mm (Thermo Scientific, Bremen, Germany). UPLC-precolumn: SeQuant ZIC-HILIC column (5-μm particles, 20 × 2 mm) (Merck, Darmstadt, Germany). UPLC column: SeQuant ZIC-HILIC column (3.5 μm particles, 100 × 2.1 mm) (Merck, Darmstadt, Germany).

Raw Data Analysis and Value Generation (in short):

LC/MS analyses were carried out in four independent experiments at 24, 48, 72, 96, and 120 h, with each value obtained from triplicate measurements. Metabolites were quantified in cell pellets and corresponding supernatants (media) under methionine-supplemented and methionine-free conditions (12 samples per time point in total). The resulting peak areas were normalized against lamivudine as an external standard. From this, the mean value and standard deviation were calculated for each triplicate. For better comparisons, the values were converted to percentages. For the values of the media, the control measurement of the medium used was defined as 100%. For the cell pellets, the highest measured value in each test series within an experiment was defined as 100%. From these values, the average mean values from the four experiments were then summarized in the individual tables. For a better overview, the results were rounded to natural numbers and shown as a heat map. The corresponding colour range is indicated individually under each table.

### 4.6. Statistical Analysis

Data collection and plotting were performed with Excel (Microsoft, Redmond, WA, USA) and GraphPad Prism (version 6.04; GraphPad Software, San Diego, CA, USA) software. Statistical analysis was performed using GraphPad Prism. Comparison between different groups was done by applying one-way ANOVA test followed by the Tukey–Kramer multiple comparison test. (ns; nonsignificant; ** *p* < 0.01; *** *p* < 0.001).

## Figures and Tables

**Figure 1 ijms-22-11630-f001:**
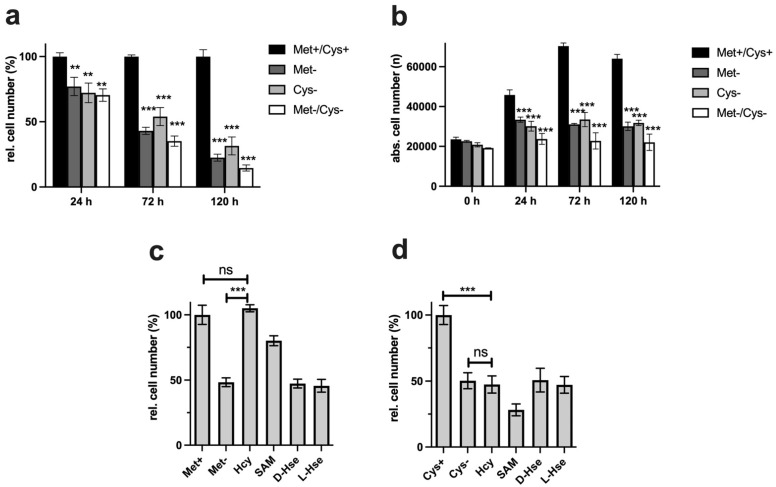
Analysis of L929 cell proliferation. A total of 10,000 cells were seeded per well, incubated overnight, and stimulated for 24, 72, and 120 h with or without methionine and/or cysteine. The proliferation of the cells was analysed by (**a**) crystal violet staining and (**b**) proliferation analysis via ImageXpress digital microscopy analysis as described in the Materials and Methods. (**a**) The figures show a summary of the results obtained from three independent experiments (five values for every group per experiment). (**b**) The figure shows one representative experiment (three values for every group per experiment). Analysis of proliferation under MetR (**c**) and CysR (**d**) and compensation with DL-homocysteine (Hcy), S-adenosylmethionine (SAM), or D-/L-homoserine (Hse). A total of 10,000 cells were seeded per well, incubated overnight, and stimulated for 72 h with or without methionine (**c**) or cysteine (**d**). For compensation, DL-homocysteine (Hcy) D-/L-homoserine (Hse) or S-adenosylmethionine (SAM) was added to the amino acid-restricted medium at a concentration of 800 µM. After 72 h, proliferation was analysed by crystal violet staining as described in the Materials and Methods. The figures show a summary of the results obtained from three independent experiments (five values for every group). The control value was set as 100%. Statistical analysis was performed using GraphPad Prism 5.0. Comparison between different groups was done by applying one-way ANOVA test followed by the Tukey–Kramer multiple comparison test. (ns; nonsignificant, ** *p* < 0.01; *** *p* < 0.001).

**Figure 2 ijms-22-11630-f002:**
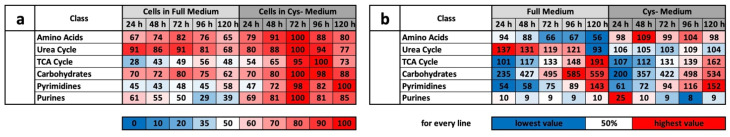
Heat map overview of the metabolites belonging to various classes and associated with pathways in cell pellets and media. The percentages of the metabolites classified as amino acids, carbohydrates, pyrimidines, and purines and belonging to the urea and TCA cycle pathways in cells (**a**) and media (**b**) were calculated as described in the Materials and Methods. In short, for cells, the highest values were set to 100%, and for medium, the blank medium values were defined as 100%.

**Figure 3 ijms-22-11630-f003:**
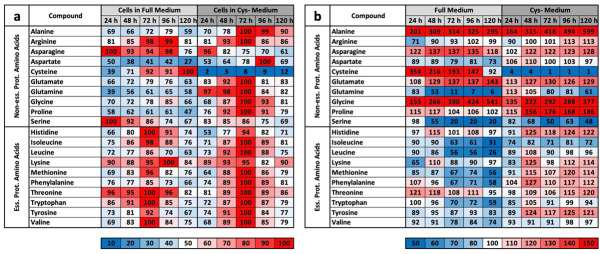
Heat map of essential and nonessential proteinogenic amino acids. LC/MS analyses were carried out in two independent experiments at 24, 48, 72, 96, and 120 h, with each value obtained from triplicate measurements. The heat map shows the summarized results. For the (**a**) cell pellets, the highest measured value in the test series was defined as 100%. For the values of the (**b**) media, the control measurement of the medium was defined as 100%.

**Figure 4 ijms-22-11630-f004:**
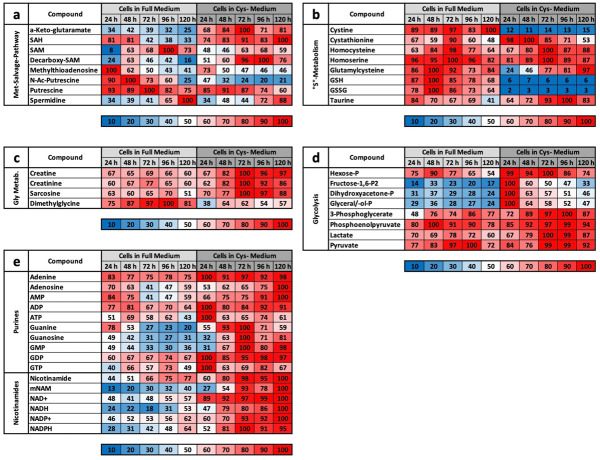
Heat map of metabolites associated with pathways/groups. LC/MS analyses were carried out in two independent experiments at 24, 48, 72, 96, and 120 h, with each value obtained from triplicate measurements. The heat map shows the summarized results: (**a**) methionine salvage pathway, (**b**) “S” (ulfur) metabolism, (**c**) Gly(cerine) metabolism, (**d**) glycolysis and (**e**) purines and nicotinamides. For the cell pellets, the highest measured value in the test series was defined as 100%.

**Figure 5 ijms-22-11630-f005:**
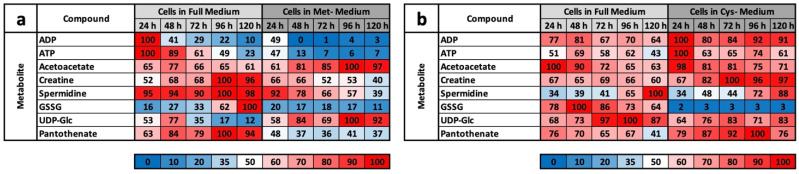
Heat map of the characteristic footprints of selected metabolites under (**a**) MetR and (**b**) CysR.

**Figure 6 ijms-22-11630-f006:**
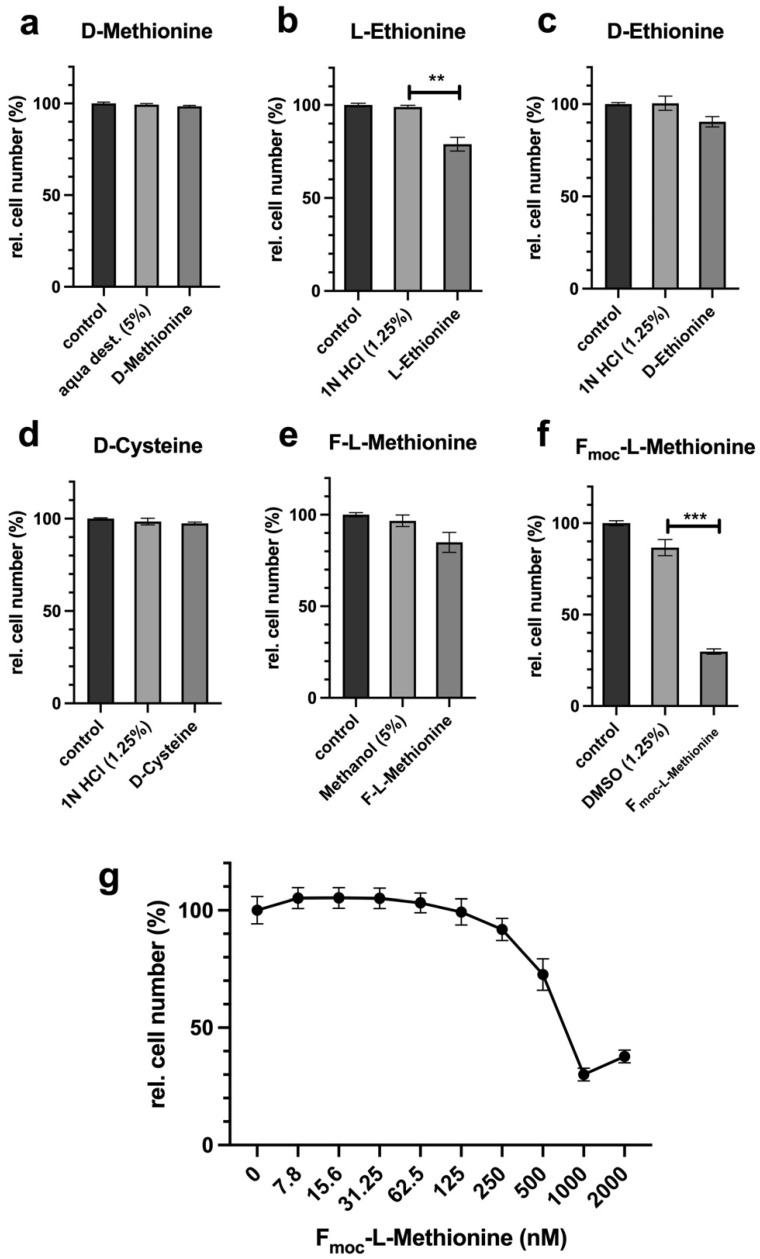
Analysis of proliferation inhibition by D-forms or potential analogues of methionine and cysteine. A total of 10,000 cells were seeded per well, incubated overnight, and stimulated for 72 h in complete medium with or without a 50× or 100× excess (values are given in brackets) of analogue or D-amino acid depending on the concentrations of methionine and cysteine and the toxicity of the solvent. The proliferation of the cells was analysed by crystal violet staining as described in the Materials and Methods. (**a**) The figures show a summary of the results obtained from two independent experiments (five values for every group per experiment). A control was full medium, and a second control was the solvent at the same concentration as in the probe with D-amino acid or analogue. (**a**) D-methionine (100×), (**b**) L-ethionine (100×), (**c**) D-ethionine (100×), (**d**) D-cysteine (50×), (**e**) formyl-L-methionine (F-L-methionine) (100×), and (**f**,**g**) Fmoc-L-methionine (100×). Comparison between different groups was done by applying one-way ANOVA test followed by the Tukey–Kramer multiple comparison test. (ns; nonsignificant, ** *p* < 0.01, *** *p* < 0.001).

**Figure 7 ijms-22-11630-f007:**
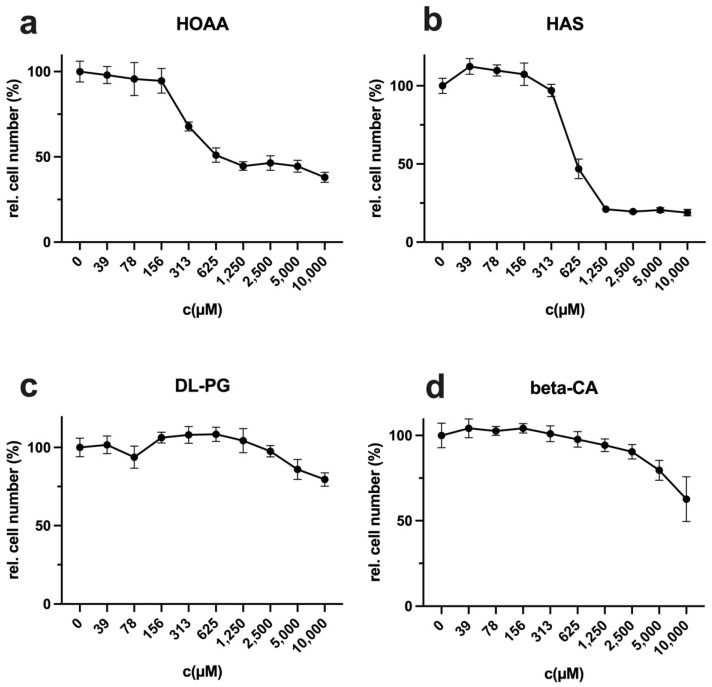
Analysis of proliferation inhibition by inhibitors of cysteine synthesis. A total of 10,000 cells were seeded per well, incubated overnight, and stimulated for 72 h in complete medium in a log2 dilution with cysteine synthase inhibitors (**a**) hydroxylamine-O-acetic acid (HOAA), (**b**) hydroxylamine solution (**c**) DL-propargylglycine (DL-PG), and (**d**) beta-cyano-L-alanine (beta-CA) with the indicated concentrations. The proliferation of the cells was analysed by crystal violet staining as described in the Materials and Methods. Control was normalized to 100%. The figures show a summary of the results obtained from three independent experiments (three values for every group).

## Data Availability

An overview of the complete results of LC/MS is added to the Appendix A.

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
