# Peer review of "Cysteine Restriction in Murine L929 Fibroblasts as an Alternative Strategy to Methionine Restriction in Cancer Therapy"

_ijms, 2021, doi:10.3390/ijms222111630_

Round 1

Reviewer 1 Report

Schmitz et al. describe Cysteine restriction in a murine model and show it as a promising therapeutic modality for cancer treatment. It is an important study that provides important insights into molecular mechanisms underlying cysteine restriction which are different from more widely studied methionine restriction approach. 

While the study is significant, there are several major gaps in the study that prevent this work to be published in its current format.

  1. It is not clear why authors selected murine L929 fibroblasts. This cell line has a normal tissue origin and may be transformed as malignant cell line, which raises ambiguity in its cell state (cancerous or normal). The results will make more impact if authors select a well characterized cancerous cell line and a control normal cell line to compare their findings and show that their findings are specific to cancer cell metabolism.
  2. Authors focus heavily on mTOR as the master regulator of metabolism. It will be interesting if mTOR inhibition leads to similar changes in the pathways studied for CysR.
  3. The analytical data is presented only as Heatmaps without any information if these changes are statistically significant. The authors mention about a Supplementary table but it was not available for reviewers to download. 
  4. Authors should add a note on the quality controls used in the LC-MS assay and the performance of external standard lamivudine.
  5. Introduction section of the manuscript is very long and descriptive almost like a review article. It will greatly help the manuscript if authors can condense it.
  6. There are several sections in the introduction which are lacking references. For example, line 52-53, 69-70, 97-98, 117-124, 157-158. Please go through the section and add references as needed.
  7. Lines 124-125: Sentence appears incomplete.
  8. Line 445: Do authors mean in vitro and in vivo?

Author Response

Dear reviewer, Unfortunately, I cannot attach the Excel file directly to the Word file. I will contact the editor so that the Excel file can be sent to you. Thank you in advance for your support, Axel Seher

Reviewer 2 Report

The manuscript on cysteine restriction and cell proliferation is interesting. There are however some issues that should be corrected.

Major The authors never mention the possibility that under cell culture conditions, autophagy can take place which will eventually, depending on the extent of deprivation, may be responsible for the cells metabolic reappraisal. The authors should consider the possibility also of an increase of cell damage in the treated cells as compared to the controls and could there be an important metabolic effect in those conditions. 

Even though the present manuscript is a continuation of previously published manuscripts with very similar conditions and analysis, the authors did not make an effort of discussing the data properly. Moreover, the introduction is extremely long which should have been shortened based on the previous recent publication. The authors insist, in both cases, on the same arguments which should have been only cited. 

The other important issue is concerning the description of LC/MS and how the authors were able to analyze part of the metabolic species referring to only percentages. Crude data should be included as a supplementary file in order to ascertain the values obtained in every case. 

In figure 4 it is unclear why in the restriction media there is an increase in glycolysis and how this glycolysis may indirectly compensate amino acid intermediates. What would happen in hypoxic conditions? 

How is the mechanism of cell toxicity by FL methionine? 

Minor events

The other of the paper should follow journal guidelines, material and methods should be after the introduction. 

There are several paragraphs that are repetitive or use however repetitively.

Author Response

(The authors gave the same response as above.)

Round 2

Reviewer 1 Report

While the authors have answered most of the questions raised in the first revision cycle, there are couple of key comments raised in the first iteration that have not been adequately answered. 

It is appreciated that authors have shared a detailed supplementary table for evaluation. However, authors have not clarified if the results shown on the HeatMaps using LC-MS data are statistically significant or not.  While the authors performed statistical analysis for other phenotypic assays described in the manuscript, it is puzzling why they did not do with LC-MS data in spite of having triplicate data points per condition.  

Authors describe using lamivudine as an external standard and to normalize the data, the external standard itself exhibited wide variability between samples. It is not clear why the peak area for the external standard would show such a high variability between samples. Taking absolute peak area (data presented in Worksheet "Data_01" of the supplementary table) of lamivudine across all the runs, there is a CV of 54% in negative mode and 48% in positive mode. Can authors explain these high CV values of their external standard. 

Reviewer 2 Report

The authors have responded most of the queries raised and the manuscript is now suitable for publication